# Home Range Estimates and Habitat Use of Siberian Flying Squirrels in South Korea

**DOI:** 10.3390/ani10081378

**Published:** 2020-08-08

**Authors:** Jong-U. Kim, Jun-Soo Kim, Jong-Hoon Jeon, Woo-Shin Lee

**Affiliations:** 1Division of Polar Life Science, Korea Polar Research Institute, Incheon 21990, Korea; 2Restoration Assessment Team, Research Center for Endangered Species, National Institute of Ecology, Yeongyang-gun, Gyeongsangbuk-do 36531, Korea; kjs88@nie.re.kr; 3Department of Forest Sciences, Seoul National University, Seoul 08826, Korea; jhstonejh@naver.com (J.-H.J.); krane@snu.ac.kr (W.-S.L.)

**Keywords:** habitat use, home range, radio telemetry, Siberian flying squirrel

## Abstract

**Simple Summary:**

The Siberian flying squirrel is the only flying squirrel species in South Korea, where it is designated a natural heritage and classed as an endangered species. The population of the species is declining worldwide throughout its distribution range. Its ecology has been studied well in different regions, especially in Finland. While several studies have been carried out on Siberian flying squirrels, little is known concerning the species’ spatial ecology in South Korea. In this study, we captured, collared, and tracked 21 animals at Mt. Baekwoon, Gangwon Province, South Korea, to investigate their movement ecology. We obtained home range size and habitat use estimates. The home range size of Siberian flying squirrels differs from those of populations in other regions. They show active movement after sunset as nocturnal species and prefer old mature deciduous forest. Our research provides valuable ecological information on this species that could help in developing management guidelines in South Korea.

**Abstract:**

Conservation measures or management guidelines must be based on species’ ecological data. The home range of the target species was studied to understand its spatial ecology, in order to protect it. The Siberian flying squirrel is the only flying squirrel species present and is considered as a protected species in South Korea. In this study, we investigated the home range, habitat use, and daily movement of Siberian flying squirrels from February 2015 to June 2016 at Mt. Baekwoon, Gangwon Province, South Korea. We tracked 21 flying squirrels using radio transmitters and analyzed the home range of 12 individuals. Flying squirrels appeared to have an overall mean home range of 18.92 ± 14.80 ha with a core area of 3.54 ha ± 3.88 ha. Movement activity peaked between 18:00–19:00 with the longest distance traveled, coinciding with sunset. In addition, we observed the preference of Siberian flying squirrels to the old deciduous forest with dense crowns. The results of the present study indicate that it is important to manage their habitat; for instance, preserving an appropriate size of mature deciduous forest is essential for Siberian flying squirrels. While our study provides needed baseline information on the spatial activity of the species, further research on topics such as the national distribution, behavior, and population dynamics of Siberian flying squirrels is needed in South Korea.

## 1. Introduction

The Siberian flying squirrel (*Pteromys volans*) is a nocturnal, herbivorous arboreal squirrel that nests in tree cavities, dreys (twig nests) and nest boxes in boreal forests. This species belongs to the Siberian fauna type and has a wide geographical distribution from western Finland to Russia (Chukotka, Sakhalin Island), Japan (Hokkaido), and the Korea peninsula in the east [1]. They mainly inhabit sheltered, spruce-dominated (*Picea abies*) mixed forests that contain a distinct deciduous tree component that provides food, as well as large aspens (*Populus tremula*) that contain nesting cavities [2,3]. The Siberian flying squirrel plays an important role as a pollinator and participates in seed dispersal in the forest ecosystem [4].

Although the Siberian flying squirrels’ population is declining worldwide due to habitat loss throughout its distribution range, this species is classed as “least concern” in the International Union for Conservation of Nature (IUCN) red list [5]. However, it has been classified as “nearly threatened” in Finland [6] and an “endangered species” in Estonia [7], and it is considered nearly extinct in Latvia. In addition, the population of Siberian flying squirrels has declined in South Korea because of habitat fragmentation and loss, as well as poor forest management practices and urban development [8]. The Siberian flying squirrel is the only flying squirrel species in South Korea. They are designated as a natural heritage by the Cultural Heritage Administration of Korea and as an endangered species by the Ministry of Environment of the Republic of Korea.

Current studies and information on this species are mainly restricted to results from Finland and Japan despite their wide distribution [9]. Various aspects of the ecology of Siberian flying squirrels have been researched, including their home range and movement patterns and habitat use [10,11,12], mating system [13], and population growth [14,15]. Studies on habitat environment [16] and nest box use [17] have been conducted, but other basic ecological data including behavior, spatial ecology, habitat utilization, and population dynamics about the species remain poorly known in South Korea despite its threatened status.

Understanding the ecological information of protected species is important to be able to make plans for conservation, but it is also essential for managing and preventing conflicts [18]. Conservation measures or management guidelines must be enforced on the basis of exact information about the target species [19]. Information on behavior, home range, and habitat use is especially important in this respect [20]. For instance, according to IUCN guidelines, information on the distribution and characteristics of habitat use is necessary for creating a suitable “habitat model” and “species distribution model” to establish a conservation plan for a target species [21,22].

We assumed that Siberian squirrels that live in South Korea might have a different home range compared with populations in other regions. For instance, the home range size of Siberian flying squirrels in Finland and Japan was different [10,12]. The size of the home range depends on the population living in different habitat environments and regions or seasons [23]. The mating system also determines the home range size and space utilization [24]. The Siberian flying squirrel is promiscuous in that both sexes copulate with more than one individual [13], and they are known to be more active during breeding season [10]. Therefore, it is difficult to directly refer to the results from other regions to establish a conservation strategy for the species in South Korea. In this context, the aim of this study was to fill in the information gap on the above-mentioned factors and provide quantitative data to establish a conservation plan for Siberian flying squirrels in South Korea. According to this purpose, we obtained estimates of the species’ home range size, movement patterns, and habitat use characteristics.

## 2. Materials and Methods

### 2.1. Ethical Statement

The Siberian flying squirrel is considered a natural heritage and an endangered species in South Korea. Procedures for capturing and handling animals were approved by the committee of Natural Heritage Division, the Cultural Heritage Administration of Korea (permit number 0456).

### 2.2. Study Area

This study was conducted at Mt. Baekwoon (37°15′43.90″ N, 127°56′20.72″ E), which is a mountainous forest located in Gangwon Province, South Korea (Figure 1). The mean annual temperature is 12.4 °C, annual rainfall is 1372 mm, and the highest elevation is 1087 m above sea level. The site comprises mixed forest habitats: Japanese red pine (*Pinus densiflora*), Japanese larch (*Larix kaempferi*), and Mongolian oak (*Quercus mongolica*) are the dominant tree species. Coniferous and deciduous trees with wide trunks are well preserved in the study area. We placed 100 nest boxes 2 to 3 m above the ground separated by a distance of about 40 m.

### 2.3. Capture and Telemetry

The field survey (capturing, collaring, and tracking) took place from February 2015 to June 2016. Siberian flying squirrels were captured from nest boxes and their sex and body mass were recorded. We attached radio transmitters (M1530 and M1540; Advanced Telemetry Systems, Inc., Isanti, MN, USA) to mature individuals [25], excluding pregnant or nursing females and juveniles. We collared animals over 85 g in body weight to avoid negative effects by the transmitter (3.7 g), not to exceed 5% of their body weight [26]. Once the radio transmitter was fitted to the neck, the flying squirrel was returned to the nest box where it was captured.

Siberian squirrels are known to be nocturnal, so radio tracking was carried out every hour from 18:00 to 06:00 according to the direction and strength of the radio signals. Collared flying squirrels were tracked for 3 consecutive days per month by using triangulation from two different stations throughout the study period. We located each individual twice within 5-min intervals to minimize any errors caused by movement. The locations of the flying squirrels were calculated using the coordinates calculated by length maximum likelihood estimators (LMLEs; 95% error ellipses) in Locate III, a radio telemetry triangulation program [27].

### 2.4. Home Range and Habitat Analysis

Home range analysis was performed using the adelhabitatHR package in R [28]. We used the minimum convex polygon (MCP) and kernel density estimator (KDE) to estimate the boundaries and sizes of the home ranges with datasets containing more than 30 fixes to calculate reliable home range estimates [29]. We used plug-in and fixed kernel methods to draw smoothing parameters in KDE [30]. For each flying squirrel, we considered 100% MCP and 95% KDE to be the home range and 50% MCP and 50% KDE to be the core area [31]. To investigate seasonal home range differences, we divided periods into breeding season (March to August) and non-breeding season (September to February) [32]. We tested the differences in the home range and core area size by sex and season with a Mann–Whitney U-test. For the distance movement of Siberian flying squirrels, we measured the distance between all consecutive locations fixed by time interval.

We used ArcGIS 10.3 (ESRI, Redlands, CA, USA) to evaluate the habitat use of flying squirrels with a forest type map [33]. The forest type map included forest type (coniferous, deciduous, and mixed), forest age, diameter at breast height (DBH), and crown density as environmental factors (Table 1). We calculated the proportions of available habitat characteristics, and also from tracked location numbers. Then we used Jacob’s index of preference to measure the preference index (PI) for the habitat selection of the Siberian flying squirrels [34]. This index provides a value range from −1 to +1, and we compared the result with categorized indication [35].

## 3. Results

A total of 21 Siberian flying squirrels (11 males and 10 females) were captured and had radio transmitters attached to them (Table 2). We continuously tracked the animals within the study area during the survey period. However, nine flying squirrels were excluded from the analysis result of their disappearance or a lack of location numbers (<30 locations). Thus, we obtained 891 locations from 12 flying squirrels (6 males, 6 females), with 32 to 186 locations per animal (Table 2).

The overall mean home range size was 15.12 ± 10.54 ha by MCP, 18.92 ± 14.80 ha by KDE, and the mean core area was 2.75 ± 3.46 ha by MCP, 3.54 ha ± 3.88 ha by KDE (Table 3, Figure 2 and Figure 3). The size of the home range (male: 16.15 ± 7.66 ha by MCP, 20.65 ± 15.30 ha by KDE; female: 14.10 ± 12.70 by MCP, 17.19 ± 14.06 ha by KDE) and core area (male: 3.70 ± 4.39 ha by MCP, 4.39 ± 5.06 ha by KDE; female: 1.80 ± 1.69 ha by MCP, 2.24 ± 1.74 ha by KDE) of males seemed slightly larger than those of females (Table 3), but there was no significant difference between sexes within the home range (MCP: *z* = −1.12, *p* = 0.26; KDE: *z* = −0.80, *p* = 0.42) and core area (MCP: *z* = −1.28, *p* = 0.20; KDE: *z* = −0.32, *p* = 0.74).

Interestingly, there were significant differences in the home range (MCP: *z* = −2.16, *p* = 0.03; KDE: *z* = −1.96, *p* = 0.04) and core area (MCP: *z* = −1.96, *p* = 0.04; KDE: *z* = −2.09, *p* = 0.03) between the breeding season and non-breeding season (Table 4). The mean home range in the breeding season (MCP: 11.10 ± 2.85 ha; KDE: 12.90 ± 2.38 ha) was larger than in the non-breeding season (MCP: 5.32 ± 3.34 ha; KDE: 5.31 ± 1.95 ha), and the core area in the breeding season was larger (MCP: 1.29 ± 0.23 ha; KDE: 2.15 ± 0.36 ha) than that in the non-breeding season (MCP: 0.53 ± 0.29 ha; KDE: 0.87 ± 0.32 ha; Table 4).

The overall mean distance traveled per hour by 12 flying squirrels was 132.75 ± 18.48 m (cumulative distance between location fixes; Figure 4). Males moved an average distance of 138.03 ± 25.62 m per hour and females moved 124.03 ± 21.18 m, but there was no significant difference between sexes (*z* = −1.96, *p* = 0.04). The most active time was around 18:00–19:00, when both males (187.71 ± 13.46 m) and females (168.74 ± 21.27 m) traveled their longest distance (Figure 4).

We characterized the habitat use patterns of Siberian flying squirrels from the home range (MCP) and 891 tracked locations (Table 5). The animals showed relatively high occurrence in deciduous forests, with 484 locations (54.37%), and in forest age class 4, with 362 locations (40.68%). We also observed high use of middle-DBH trees, with 494 locations (55.43%), and dense crown areas, with 529 locations. There was a low proportion of habitat use in small DBH trees, with 131 locations (14.72%), and sparse crown areas, with 33 locations (3.65%). Jacob’s index suggests that the flying squirrels showed a strong preference for age class 6 (PI = 0.82) and moderate preferences for deciduous forest (PI = 0.21), age class 3 (PI = 0.59), large-DBH trees (PI = 0.32), and dense crown areas (PI = 0.21; Table 5).

## 4. Discussions

The mean home range size of Siberian flying squirrels was 15.12 ha (MCP: range 1.88–38.19 ha) and 18.92 ha (KDE: range 3.78–52.69 ha) at Mt. Baekwoon, Gangwon Province, South Korea. The results of our study differ from those of previous studies. Hanski et al. [10] reported a mean home range size of 34.10 ha (MCP: range 2.70–132.00 ha) in Finland, which is relatively larger than the home range of our study. However, a study from Japan [12] showed a home range of 4.14 ha (MCP: range 1.10–16.60 ha), which is smaller than the home range in South Korea. The home range size differs for populations living in different habitats or regions [23]. The size of our study area was 20 ha, whereas the study areas in Finland ranged from a few hundred to thousands of hectares, covering much more area compared with our study [10]. Additionally, the sizes of the study areas in Japan were much smaller than ours, ranging from 2.1 to 13.0 ha. Furthermore, the main tree species were different in South Korea, Finland, and Japan [12]. Therefore, the described intraspecific differences in home range size might depend on an environmental habitat gap across its geographic range. Other arboreal squirrels—American red squirrel (*Tamiasciurus hudsonicus*), gray squirrel (*Sciurus carolinensis*), fox squirrel (*S. niger*), and Eurasian red squirrel (*S. vulgaris*)—also showed a positive relationship between the home range and forest size [36,37].

The results of our study are similar to those of other studies showing that flying squirrels had a larger home range during breeding season. In general, males usually appear to have a larger home range than females [38], which is also true of Siberian flying squirrels; males enlarge their home range to enhance their chance of mating with many females [10,12,39]. Contrary to this, our study did not observe differences in home range or daily movement between males and females. This result may be affected by the large individual variation and small sample size [1].

Nevertheless, there are some possible explanations that could account for the results of our study. First, males will have more chances to have access to females in a high-density population, so the home range size of males should be smaller than that with a low-density population [38]. Second, females increase their movement to secure better nesting sites and compete with each other [39]. Considering that Siberian flying squirrels appear to be opportunistic, it is possible that they have different home ranges depending on habitat conditions and resource availability [40]. In addition, the density of many arboreal squirrel species tends to increase with habitat loss, such as forest fragmentation, and hence, density increases in smaller habitats [41]. In the present study, there seemed to have been abundant nesting sites because of the nest boxes placed at Mt. Baekwoon; this may explain the differences in home ranges and movements among regions. However, additional studies are required to clarify the specific spatial ecology of Siberian squirrels in South Korea, such as gender-based differences in sexual home range size, home range overlap, and the relationship between the home range and habitat composition, with intensive radio telemetry surveys.

The animals were the most active and traveled the longest distance after sunset and did not move after sunrise. This confirms their ecological characteristic as a nocturnal species [42]. In our study, flying squirrels preferred deciduous forests despite a higher proportion of coniferous forests, and trees with old thick trunks and dense crowns. These preferences may reflect their ecological characteristics as an arboreal squirrel. The occurrence of Siberian flying squirrels increases with forest age and the volume of spruce trees and deciduous trees, and these mature forests are associated with foraging habitats and nesting sites for the species [43]. These well-preserved forests can offer benefits to animals, providing more cavities [10] and higher trees for launching and gliding locomotion [44], and the dense canopy can protect them from predators [45].

We found that the home range size of Siberian flying squirrels at Mt. Baekwoon differed from those of other populations in different regions, and these results may be caused by the habitat productivity status [9]. However, habitat preference was similar to that of other Siberian flying squirrels [46]. The results of the present study indicate that it is important to manage the habitat; for instance, preserving an appropriate size of mature deciduous forest is essential for Siberian flying squirrels. In addition, we suggest placing artificial nest boxes to increase the density of flying squirrels and their reproduction, because of the lack of large trees in South Korea [17]. While our study provides needed baseline information on the spatial activity of species in South Korea with a limited sample, some parallel and further studies are needed. Future research would need to account for discrepancies in habitat composition, such as the forest structure where these squirrels live and where they do not. Furthermore, the national distribution of the species and its dispersal behavior and gliding locomotion should be investigated to understand the population dynamics of Siberian flying squirrels in South Korea.

## 5. Conclusions

We investigated the home range and habitat use of multiple Siberian squirrels at Mt. Baekwoon and provide new quantitative data on local movement patterns. This study reveals the home ranges and habitat use of the Siberian flying squirrel population living in South Korea for the first time. Furthermore, the results of this study will be helpful in providing novel information on the actual home range and habitat preference that might help in the development of management guidelines and conservation strategies for the species in South Korea.

## Figures and Tables

**Figure 1 animals-10-01378-f001:**
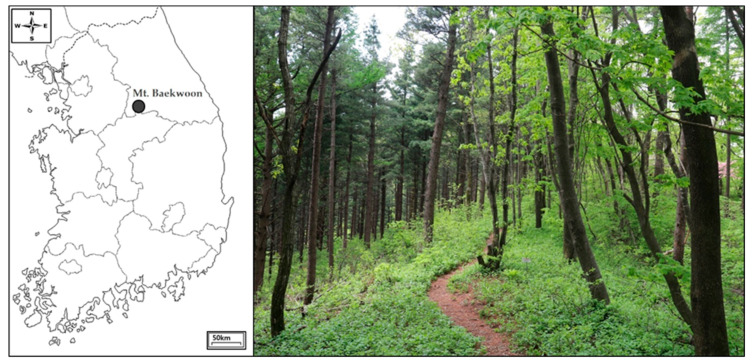
Location of Mt. Baekwoon in Gangwon Province, South Korea (37°15′43.90″ N, 127°56′20.72″ E).

**Figure 2 animals-10-01378-f002:**
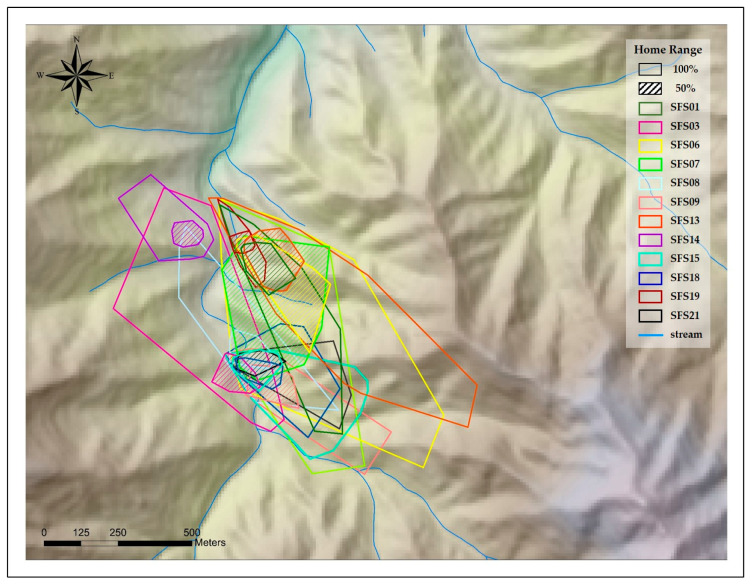
Home range size and core area of Siberian flying squirrels at Mt. Baekwoon, Gangwon Province, South Korea, based on the minimum convex polygon (MCP).

**Figure 3 animals-10-01378-f003:**
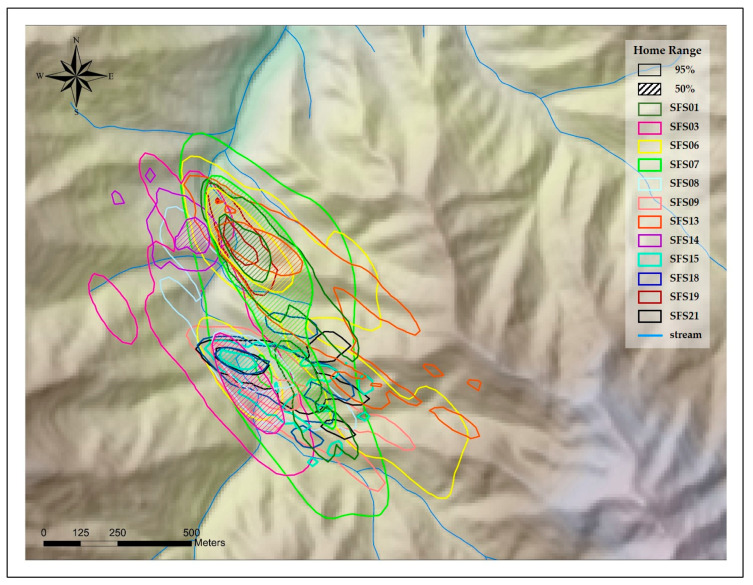
Home range size and core area of Siberian flying squirrels at Mt. Baekwoon, Gangwon Province, South Korea, based on the kernel density estimator (KDE).

**Figure 4 animals-10-01378-f004:**
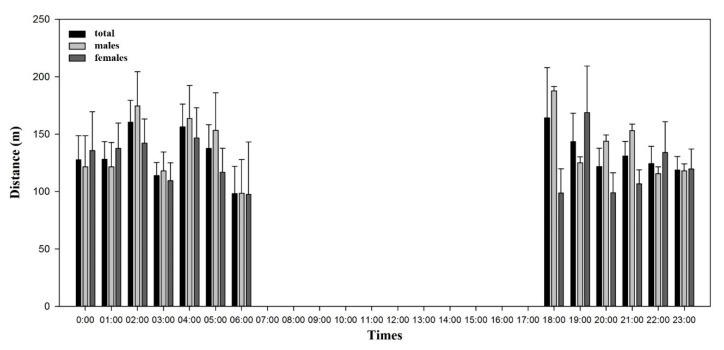
Daily movement of Siberian flying squirrels at Mt. Baekwoon, Gangwon Province, South Korea.

**Table 1 animals-10-01378-t001:** Classification and description of environmental factors.

Environmental Factor	Class	Description
Forest age	1	At least 50% of total canopy cover is occupied by 1 to 10 year-old trees
	2	At least 50% of total canopy cover is occupied by 11 to 20 year-old trees
	3	At least 50% of total canopy cover is occupied by 21 to 30 year-old trees
	4	At least 50% of total canopy cover is occupied by 31 to 40 year-old trees
	5	At least 50% of total canopy cover is occupied by 41 to 50 year-old trees
	6	At least 50% of total canopy cover is occupied by 51 to 60 year-old trees
	7	At least 50% of total canopy cover is occupied by 61 to 70 year-old trees
	8	At least 50% of total canopy cover is occupied by 71 to 80 year-old trees
	9	At least 50% of total canopy cover is occupied by 81 to 90 year-old trees
DBH	1	Saplings: trees with less than 6 cm DBH occupy at least 50% of total canopy cover
	2	Small: trees with 6–18 cm DBH occupy at least 50% of total canopy cover
	3	Middle: trees with 19–30 cm DBH occupy at least 50% of total canopy cover
	4	Large: trees with larger than 30 cm DBH occupy at least 50% of total canopy cover
Crown density	1	Sparse: dominant tree canopy occupies at least 50% of total canopy cover
	2	Moderate: dominant tree canopy occupies at least 51–70% of total canopy cover
	3	Dense: dominant tree canopy occupies more than 70% of total canopy cover

DBH, diameter at breast height.

**Table 2 animals-10-01378-t002:** Technical details, ID, captured date, sex, weight, number of locations, and duration of telemetry of Siberian flying squirrels at Mt. Baekwoon, Gangwon Province, South Korea.

ID	Captured Date	Sex	Weight (g)	Number of Locations	Duration of Telemetry
SFS01	21 February 2015	Male	109	42	3 months
SFS02	28 May 2015	Male	91	Lost	–
SFS03	23 April 2015	Female	101.5	32	2 months
SFS04	24 April 2015	Female	98.8	Lost	–
SFS05	26 June 2015	Female	111.5	12 (excluded)	2 months
SFS06	22 August 2015	Female	89	56	5 months
SFS07	25 September 2015	Male	99	34	4 months
SFS08	19 October 2015	Male	119	38	3 months
SFS09	19 October 2015	Female	91.5	44	3 months
SFS10	19 October 2015	Male	99.5	Lost	–
SFS11	21 December 2015	Female	91.5	26 (excluded)	2 months
SFS12	22 December 2015	Male	86.8	10 (excluded)	1 month
SFS13	24 April 2015	Male	103	186	10 months
SFS14	26 September 2015	Female	104	85	7 months
SFS15	21 February 2015	Male	102	154	8 months
SFS16	24 February 2016	Female	124	16 (excluded)	1 month
SFS17	25 June 2015	Male	106	Lost	–
SFS18	22 August 2015	Male	89.4	143	8 months
SFS19	30 March 2016	Female	119.5	36	2 months
SFS20	26 February 2016	Male	97.1	15 (excluded)	2 months
SFS21	30 March 2016	Female	91	41	3 months

**Table 3 animals-10-01378-t003:** Estimates of the home range and core area of Siberian flying squirrels at Mt. Baekwoon, Gangwon Province, South Korea.

ID	Sex	Minimum Convex Polygon (ha)		Kernel Density Estimator (ha)	
		100%	50%	95%	50%
SFS01	M	12.76	1.95	15.49	2.29
SFS07	M	27.62	13.43	52.69	15.46
SFS08	M	11.93	1.37	18.63	3.65
SFS13	M	25.9	2.89	22.05	3.13
SFS15	M	10.64	1.46	6.53	0.62
SFS18	M	8.02	1.12	8.54	1.17
Mean ± SD		16.15 ± 7.66	3.70 ± 4.39	20.65 ± 15.30	4.39 ± 5.06
SFS03	F	23.17	1.42	30.72	4.24
SFS06	F	38.19	5.44	41.39	5.49
SFS09	F	9.40	1.80	14.06	3.12
SFS14	F	5.27	0.75	5.27	1.11
SFS19	F	1.88	0.46	3.78	1.09
SFS21	F	6.67	0.93	7.89	1.08
Mean ± SD		14.10 ± 12.70	1.80 ± 1.69	17.19 ± 14.06	2.24 ± 1.74
Total mean ± SD		15.12 ± 10.54	2.75 ± 3.46	18.92 ± 14.80	3.54 ± 3.88

**Table 4 animals-10-01378-t004:** Differences in the home range and core area of Siberian flying squirrels between breeding and non-breeding seasons at Mt. Baekwoon, Gangwon Province, South Korea.

	Minimum Convex Polygon (ha)		Kernel Density Estimator (ha)	
	100%	50%	95%	50%
Breeding season	11.10 ± 2.85 *	1.29 ± 0.23	12.90 ± 2.38	2.15 ± 0.36
Non-breeding season	5.32 ± 3.34	0.53 ± 0.29	5.31 ± 1.95	0.87 ± 0.32
Z	−2.16	−1.96	−1.96	−2.09
*p*	0.03	0.04	0.04	0.03

* Mean ± SD.

**Table 5 animals-10-01378-t005:** Proportion of available habitat characteristics, proportion of habitat use from tracked locations, and Jacob’s preference index (PI) of Siberian flying squirrels by environmental factors at Mt. Baekwoon, Gangwon Province, South Korea.

Environmental Factor		Proportion of Available Habitat Characteristics (%)	Proportion of Habitat Use from Tracked Locations (%)	PI
Forest type	Coniferous	48.29	29.85	–0.24
	Deciduous	35.72	54.37	0.21
	Mixed	15.99	15.78	–0.01
Forest age	Class 2	9.27	0.00	–1.00
	Class 3	7.76	29.86	0.59
	Class 4	35.13	40.68	0.07
	Class 5	47.36	24.64	–0.32
	Class 6	0.48	4.82	0.82
DBH	Small	11.06	14.72	0.14
	Middle	73.43	55.43	–0.14
	Large	15.51	29.85	0.32
Crown density	Sparse	11.62	3.65	–0.52
	Moderate	49.83	37.03	–0.15
	Dense	38.55	59.32	0.21

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
