# Peer review of "Home Range Estimates and Habitat Use of Siberian Flying Squirrels in South Korea"

_animals, 2020, doi:10.3390/ani10081378_

Round 1
Reviewer 1 Report
Here I have reviewed the manuscript titled ‘Home Range Estimates and Habitat Use of Siberian Flying Squirrels in South Korea.’ The authors collected information about the habitat characteristics, home range and active time of flying squirrels in South Korea, a species that is under conservation management. To fill in this information, the authors monitored squirrels using radio transmitter and conducted regular field survey. I agree with the authors that ecological information about a species is important for informing local conservation managements. The authors also did a good job in comparing the similarities and differences of flying squirrels residing in different countries - these information help the readers to obtain a more global perspective in understand whether any of the studied variables may have an effect on the squirrels.
Overall, I think the study is interesting, though quite narrow. If it has to become publishable, it needs more work for example, in analyses, adding/reorganising information for the study background (see below for details).
- it appears to me that the authors compared sex differences in home range, movement patterns and active time. However, no information related to sex is provided for readers.
- Do the authors expected to find a difference in sex and season in relation to home range size, habitat use and active pattern? If so, please state clearly in the introduction.
- How do studying these varying matter to the local and global conversation managements? The authors did provide the ‘what’, but missing the ‘how’.
- Why did the authors did not take both variables, ‘sex’ and ‘season’, in the same model to predict habitat use, active patterns etc?
- I am not understanding this very well. The authors mentioned they have excluded 9 individuals for analyses. However, when looking at the table, these 9 individuals carried the radio transmitter for 1-2 months. As the authors checked on them for three consecutive days per month, there should be some data about home range size, habitat use and not least, activity pattern (depending on the season that the authors captured them). So why would the authors excluded these 9 individuals for analyses? Please clarify.
Abstract
Line 36 – the sentence ‘further research is needed’ is not helpful to understand anything. A suggestion is to think about how future research should build on the provided baseline information so that conversation managements could be done properly for the flying squirrels.
Introduction
The authors should also provide a brief information about the study species, in particular if their interests included sex difference in home range size, habitat use and activity pattern (I don’t get any of these information until I read the Discussion).
Line 49-56 this para is a bit messy and should be reorganised. First, there is contradicted information - ‘least concern’ vs. ‘nearly threatened’. Please revise the sentence to something like ‘However, the population of Siberian flying squirrels is declining worldwide due to habitat loss. While this species is considered as ‘least concern’ by the IUCN, it has been classified as ‘nearly threatened’ in Finland, ‘endangered species’ in Estonia, and ‘nearly extinct’ in Latvia. Second, populations of flying squirrels in South Korea appear to have similar trend (declining) to countries in Europe. A sentence summarising such information will tell the readers about the situation in South Korea.
Line 61 like what basic ecological data is missing for this species? Would the authors name a few that are related to the current study?
Line 66-67 missing linkage - I agree with the authors that having the basic information about a species is important for conservation. However, how would such information be useful for conservation management? Please elaborate the idea.
Line 69 wrong clause. Why ‘furthermore’? please revise the topic sentence.
Materials and Methods
Ethical statement
Line 79-83. Redundant and missing information. The sentence ‘There is an ….. on laboratory animals and animal testing’ is redundant, please delete it. If it has to be kept, then please find the relevance to the current study procedures. The key information that should be provided for the ethical statement is what authority approved the procedures for the study, which the authors have provided (line 80-83) and there is any animal got injured (or death) due to the capture and handling procedures. Please elaborate.
Capture and Telemetry
Line 99. How do the authors differentiate individuals are mature?
Line 100. What is considered as ‘heavy enough’ – what is the average weight of a mature flying squirrel?
Line 101. The capture and handling procedure may be familiarised with the authors, but if a reader would like to conduct the same study, one would like to know whether these collars each has a unique number or emit a different code so that the number of individuals in an area could be estimated. I would also need more information about how the authors track the squirrels – did the authors walked around the forest during each check? Or just stayed next to a nest box for 15 minutes? Or just stayed at a central location? These more detail procedure is important for replication and subsequent monitoring.
Results
Line 131-132. Why excluded the 9 squirrels that may have provided data (as these squirrels carried the transmitter for 1-2 months and that the authors checked the area for 3 consecutive days per month.
Line 136-139 as said, why did the authors did not use a model that included ‘sex’ and ‘season’ for analysis?
Table 2 is a bit confusing. For example, the authors indicated there were 11M and 10F but the Total* is representing Mean and SD. Could the authors find a way to place Mean± SD in the table?
Line 162-167 – did habitat use differ in males and females or breeding and non-breeding season too?
Discussion
Line 186-190. Information about mating system and the importance in relation to conversation managements should be placed much earlier, favourably in the introduction – so that the authors could pave a way to explain ‘sex’ would be a variable of interest.
Author Response
Thnk you very much for comments and suggestions.
Please see the attachment.

Reviewer 2 Report
This paper describes for the first time in South Korea the results of a radio telemetry study on the Siberian flying squirrel. The authors report on the home-range sizes and daily movement patterns of both sexes, as well as seasonal variations and habitat use. Overall the authors present a convincing and important study on this species in the Korean peninsula for the first time. The data generated in this study will be useful in developing conservation strategies for this declining species in Korea. It is especially useful to be able to generate new ecological data on such a widespread species which would be expected to display significant variations in its home-range size, movement patterns, population densities and habitat use in varying ecological conditions.
I have no major criticisms and have highlighted numerous very minor grammatical changes to improve the flow of the paper. I have highlighted these on the paper.
The discussion does perhaps need a bit more development. It is not clear whether other studies on this species used similar estimators for home-range size or not, which could affect comparisons. I think also the discussion would benefit from a more comprehensive and nuanced discussion of the interplay between home-range size, population density, habitat extent and quality, and proportion of physical characteristics that allow gliding locomotion and provide nest sites and havens from predators. All of these factors presumably vary between the different published studies. For example, the authors recommend leaving areas of old-growth forest for flying squirrels, but surely also the pattern of distribution of these patches will be important, i.e. one big patch, several patches close to or far apart from each other? These are important considerations for developing conservation plans for the species. Also some indication of minimum tree age or height/width that encourages flying squirrel occupation and the possible impact of climate change would be useful. Clearly, given the species' ability to adapt to differing ecological conditions it displays sufficient plasticity in its behaviour and ecology that this study highlights the need for numerous parallel studies across Korea if a national conservation plan is to be developed. Although this study is very useful, it would be wrong to depend on it nationally or regionally.
Author Response

(The authors gave the same response as above.)

Reviewer 3 Report
I have provided extensive comments and suggestions in the attached manuscript.
I believe that there is a need to provide maps showing the distribution of the different habitat types as well as the boundaries of the study area and of the reserve/area of suitable habitat. It should be possible to overlay the home range isopleths over such a base map.
In order to show habitat selection it is necessary to know the amount of each habitat class in the area so that this can be compared with the amount of use in order to determine true habitat selection.
The paper requires a more detailed description of the study area including a topographic map which might explain the similarity in shape and directionality of the home ranges.

Author Response

(The authors gave the same response as above.)

Round 2
Reviewer 1 Report
I have reviewed the first version of the manuscript. In this revised version, the authors did a good job in improving the manuscript by addressing the reviewer’s comments, which I do not find major concerns. However, there is also a comment that I raised up previously have not been addressed properly (this is minor and see below). Most of my comments for this version are about tying up information across the manuscript.
Line 20 In simple summary, the authors mentioned the squirrels’ home range size is different from those reported. This is followed by the next sentence about their active period and habitat preference, but the authors did not mention whether these characteristics are also similar to those squirrels residing at different regions. Would the authors add some information here?
Abstract
Line 30 the sample size of 21 was true, but as the results are based on 12 squirrels, it becomes misleading. The authors may think about revising this sentence.
Introduction
Line 50 – grammatical issue. This is an issue raised previously. The ‘although’ is misused or not followed by a contrasting information. Please revise the sentence to ‘Although Siberian flying squirrels’ population is declining worldwide due to habitat loss throughout its distribution range, this species is classed as ‘least concern’.
Line 72 ‘distributing’?
Line 73 linkage between assumption and argument. The authors assumed that the home rage of the squirrels would be different in other regions. Although the authors have provided what the factors that affect the size of home range, they may want to add the argument that squirrels residing in Finland and Japan have different home range size before ‘the size of home range depends on….’
Author Response
Thank you very much for your comments and suggestions.
Please see the attached file.

Reviewer 3 Report
I have made comments in the attached manuscript. My previous comments have been adequately addressed.
There seems to me to be enough information her for a brief analysis and discussion of the overlap of the core home range areas. The core areas seem to be concentrated in two locations and some additional description of the characteristics of these two areas might be useful.

Author Response

(The authors gave the same response as above.)
